# The Current Molecular and Cellular Landscape of Chronic Obstructive Pulmonary Disease (COPD): A Review of Therapies and Efforts towards Personalized Treatment

**DOI:** 10.3390/proteomes12030023

**Published:** 2024-08-16

**Authors:** Luke A. Farrell, Matthew B. O’Rourke, Matthew P. Padula, Fernando Souza-Fonseca-Guimaraes, Gaetano Caramori, Peter A. B. Wark, Shymali C. Dharmage, Phillip M. Hansbro

**Affiliations:** 1School of Life Sciences, Faculty of Science, University of Technology Sydney, Centre for Inflammation, Ultimo, NSW 2007, Australia; luke.farrell@student.uts.edu.au; 2School of Life Sciences, Faculty of Science, University of Technology Sydney, Ultimo, NSW 2007, Australia; matthew.padula@uts.edu.au; 3Frazer Institute, Faculty of Medicine, The University of Queensland, Woolloongabba, QLD 4102, Australia; f.guimaraes@uq.edu.au; 4Pulmonology, Department of Medicine and Surgery, University of Parma, 43126 Parma, Italy; gaetano.caramori@unipr.it; 5School of Translational Medicine, Monash University, Melbourne, VIC 3000, Australia; peter.wark@monash.edu; 6Centre for Epidemiology and Biostatistics, School of Population and Global Health, The University of Melbourne, Melbourne, VIC 3000, Australia; s.dharmage@unimelb.edu.au

**Keywords:** COPD, emphysema, proteomics, pathogenesis, treatment discovery and development

## Abstract

Chronic obstructive pulmonary disease (COPD) ranks as the third leading cause of global illness and mortality. It is commonly triggered by exposure to respiratory irritants like cigarette smoke or biofuel pollutants. This multifaceted condition manifests through an array of symptoms and lung irregularities, characterized by chronic inflammation and reduced lung function. Present therapies primarily rely on maintenance medications to alleviate symptoms, but fall short in impeding disease advancement. COPD’s diverse nature, influenced by various phenotypes, complicates diagnosis, necessitating precise molecular characterization. Omics-driven methodologies, including biomarker identification and therapeutic target exploration, offer a promising avenue for addressing COPD’s complexity. This analysis underscores the critical necessity of improving molecular profiling to deepen our comprehension of COPD and identify potential therapeutic targets. Moreover, it advocates for tailoring treatment strategies to individual phenotypes. Through comprehensive exploration-based molecular characterization and the adoption of personalized methodologies, innovative treatments may emerge that are capable of altering the trajectory of COPD, instilling optimism for efficacious disease-modifying interventions.

## 1. Introduction

Chronic obstructive pulmonary disease (COPD) is a chronic inflammatory disease that impacts the respiratory system [1,2]. It is also a systemic disease that has important bidirectional interactions with other organs such as the gut (gut–lung axis) heart, spleen, liver, bone marrow, thymus, central nervous system (brain–lung axis), and skeletal muscle. The Global Initiative for Chronic Obstructive Lung Disease (GOLD) defines COPD as “a heterogeneous lung condition characterized by chronic respiratory symptoms (dyspnoea, cough, sputum production, exacerbations) due to abnormalities of the airways (bronchitis, bronchiolitis) and/or alveoli (emphysema) that cause persistent, often progressive airflow obstruction” (Global Initiative for Chronic Obstructive Lung Disease, 2023). According to the GOLD report, spirometry readings post-bronchodilation that report a FEV1/FVC < 0.7 confirm the diagnosis of COPD [3]. Part of the difficulty in diagnosing COPD is the overlap it shares with other respiratory diseases such as asthma, idiopathic pulmonary fibrosis (IPF), and cystic fibrosis (CF) [4,5]. Characteristics such as inflammation, airway remodeling, persistent coughing, and dyspnoea are common symptoms shared by these diseases (Table 1). Specifically, COPD and asthma are difficult diseases to separate; this is due to them sharing a range of similar biological and clinical manifestations, such as chronic inflammation and airflow obstruction [3,6]. The specific features of COPD include fixed airflow obstruction and emphysema. The diagnosis of COPD can be performed using techniques such as spirometry, computed tomography (CT) scans and plethysmography techniques [3,6,7]. While techniques such as spirometry are generally the first line of diagnosis for COPD, it is important to recognize that some patients with early COPD will present normal lung function with spirometry. Other, more expensive methods such as CT scanning and plethysmography can detect patients who would otherwise not present with early COPD [8]. The concept of COPD endotypes is introduced to help decipher the complexity of COPD; the key endotypes proposed include the endotypes involved in airway remodeling, neutrophilic inflammation, the microbiome, eosinophilic inflammation, and increased apoptosis paired with decreased repair [9].

## 2. The Development of COPD

The development of COPD is influenced by a wide range of risk factors. The most significant risk factor for COPD is cigarette smoke (including second-hand smoking) with ~1.5 of the 3 million deaths from COPD globally each year being attributed to smoking [23]. Eight million die from smoking-induced diseases each year [24]. The other significant COPD risk factors associated with increased mortality include: ambient particulate matter pollution, occupational particulate matter (gases and fumes), ambient ozone pollution, household air pollution from solid fuels, and lead exposure [23]. Other risk factors are present at birth, such as genetic factors, or in the womb upon exposure to pollutants/chemicals [25]. Indeed, low lung function trajectories are strongly linked to the development of COPD and its evolution from pre- to progressive COPD [26,27,28]. It is important to note that when, compared against mortality statistics for 2007, every risk factor category shows an increase, with the only exception being household air pollution from solid fuels [23].

Smoking rates are declining in some developed countries but continue to increase in others. The number of smokers is rising in African, Eastern Mediterranean and south east regions and is decreasing in the Americas, Europe, and the Western Pacific regions [29]. Globally, COPD prevalence continues to increase, and it has progressed from the 5th to the 3rd most common cause of illness and death globally. Furthermore, air pollution in many regions is increasing at an alarming rate, as is vaping (the use of E-cigarettes). This is particularly prevalent in adolescents [30,31]. Some regions see >40% of children between 15 and 16 using vapes or E-cigarettes [29]. While the long-term impact of E-cigarettes is currently largely unknown, studies indicate that chronic exposure alters lung physiology, with pathogenesis mechanisms very similar to COPD [32]. Alarmingly, there is a significant association between E-cigarette usage and COPD diagnosis, even in patients who never smoked cigarettes [33]. Furthermore, other COPD risk factors, such as ambient air pollution and asthma, have increased in the past couple of decades [34].

Notably, even with risk factors being present from an early age, in a large portion of people who develop COPD, diagnosis usually does not occur until age 40–50 [25,35]. However, late diagnosis does not mean that there were no indicators of early COPD, such as increased resistance or impaired reactance. Late diagnosis may be because lung function must decline substantially before the sufferer takes action and employs the use of spirometry to determine the existence of COPD-related lung obstruction [36]. Spirometry is the primary diagnostic method and its use can result in false negatives in patients with early-stage COPD, who may benefit the most from early intervention [37,38,39]. Symptomatic patients may have spirometry results that do not demonstrate airflow obstruction, and therefore fail to meet the physiological criteria for COPD as defined by GOLD, but do demonstrate abnormal lung function, such as increased resistance or impaired reactance, when assessed by impulse oscillometry or when emphysema is detected via high-resolution CT [37,39,40]. These other techniques can detect COPD in ways that spirometry cannot since a key feature of COPD is airway remodeling; however, these are expensive and require a significantly higher level of expertise regarding the operator and those assessing the outcomes.

Other key risk factors that increase the chance of an individual developing COPD include asthma and respiratory infections, with a significant increase if these are present early in life [41]. For a more in-depth review on the physiological changes occurring in COPD and current methods of diagnosis, see [3,42,43].

## 3. Chronic Inflammation and Bronchitis in COPD

A common theme in COPD pathogenesis is pre-COPD exposure to an irritant that induces airway and lung inflammation, e.g., cigarette smoke or biofuel gas/particulates from burning fuel (firewood, dung cakes, crop residues, coal, kerosene) [44,45,46,47,48,49,50]. Chronic exposure to these irritants is strongly associated with the pathogenesis and the progression of COPD [51,52,53]. This is exacerbated by respiratory infections, which happen more frequently in patients with COPD, creating a vicious cycle [45,54,55,56,57,58,59,60]. Acute inflammation does not appear to cause structural damage to the same extent that prolonged chronic inflammation and bronchitis do. Acute inflammation is associated with positive outcomes, with most cases ending with full recovery and a return to tissue homeostasis, provided the inflammatory stimulus is removed and the immune cells are correctly functioning [61,62]. In contrast, chronic inflammation drives constant rounds of wounding and repair that over time overwhelm the protective systems, resulting in neutrophil-mediated destruction through the secretion of proteases and pro-inflammatory molecules, tissue remodeling, and emphysema [61]. While neutrophils are a necessary and critical part of immune defence, their increased abundance is associated with tissue damage and increased inflammatory responses [63]. We recently showed that there are five different phenotypes of circulating neutrophils that also occur in the lungs in COPD, but only one is associated with disease features. Recently we, and many others, also made critical advances in investigating the immune responses crucial for driving chronic inflammation in COPD, which have become important therapeutic targets (see Figure 1) [45,54,55,56,57,58,59,60,64]. These include cell-derived responses from mast cells and their proteases, T and B; natural killer (NK) and innate lymphoid cells; macrophages; and neutrophils [45,58,65,66,67,68,69,70]. They also include molecular-based responses involving inflammasomes; oxidative stress; IL-22; TRAIL; micro(mi)RNAs and the pathways they control; stress granules; and amine and lysyl oxidases [70,71,72,73,74,75,76,77,78,79,80]. Finally, chronic inflammation is also influenced by the lung and gut microbiomes and the gut–lung axis [81,82,83,84,85,86,87].

A crucial step in returning to tissue homeostasis from inflammation is the activation of resolving pathways and the effective clearance of apoptotic neutrophils [88,89,90]. If these cells are not cleared, they undergo secondary necrosis and necroptosis, characterized by the loss of membrane integrity [91]. This further feeds the inflammatory response and results in further tissue damage as the necrotic neutrophils release cytotoxic oxidants, proteases, damage-associated molecular patterns (DAMPs), and pro-inflammatory molecules [92]. Specifically, in COPD, one of the proteases released is neutrophil elastase (NE), which decreases the integrity of the alveolar structure through the modification of the extra cellular matrix (ECM), which directly results in the development of emphysema [93]. Furthermore, in some respiratory diseases such as CF or bronchiectasis, NE cleaves cell markers that direct macrophages to clear apoptotic cells; this further increases the burden of cells undergoing necrosis in the diseased lung, resulting in further inflammatory cytotoxin release [93].

## 4. Chronic Structural Changes

Chronic inflammation in bronchitis is strongly associated with and likely leads to bronchial and parenchyma remodeling and emphysema [94,95]. Airway remodeling and emphysema eventually lead to a significant decrease in FEV_1_/FVC ratio. This reaches the threshold of COPD GOLD stage 1, which is detectible by spirometry, and reduces the gas exchange detectable by assessing the diffusing capacity for carbon monoxide (DL_CO_). Nevertheless, smokers who have acceptable FEV_1_/FVC ratios can still present with COPD symptoms; remodeling and emphysema, which are only detectible using CT scanning, or parametric response mapping (PRM) [96]. Indeed, 25–30% of lung tissue may be affected by emphysema without impacting the FEV_1_/FVC ratio [97], and these individuals likely suffer from early or pre-COPD [96,98].

Airway thickening is also strongly associated with COPD through multiple pathways and is heavily influenced by inflammation [99]. The most common mechanism by which this takes place is the hypertrophy of airway smooth muscle (ASM), further narrowing the airways and reducing airflow [100]. The ECM has a critical role in airway thickening, with only 75% of the ASM layer actually consisting of smooth muscle cells and the rest being ECM, mast cells, and blood vessels [101]. The ECM, containing a multitude of different molecules such as collagen, fibronectin, fibulin, tenascin, proteoglycans and elastic fibers contributes to multiple different pathways in airway thickening in ASM, with the composition influencing stiffness, growth, and structure [101,102]. The increased deposition of collagen and fibrosis of the airways in COPD further narrows the airways [103]. We have shown the critical pathogenic roles of fibulin-1c in airway remodeling, and of emphysema in COPD, whereby it acts as a scaffold other ECM proteins can build around to promote fibrosis [104].

Emphysema is a major disease feature that often occurs in patients with COPD and it is characterized by the destruction of the ECM and tissues and the enlargement of distal airspaces [105,106,107]. This leads to the rupture of alveolar walls that then form one large air pocket, reducing the overall surface area–volume ratio and the capacity of the lungs to effectively perform gas exchange [108,109,110,111]. This tissue destruction, the loss of the ECM, and the enlargement of alveoli reduce the elasticity of the lung. This loss of elastic recoil leads to airway collapse and a reduced FEV_1_:FVC ratio [112]. We have also recently demonstrated the critical role of increased TLR7 responses in COPD, which drive the destruction of alveolar epithelial cells that promote emphysema [65].

Cigarette smoke can change and modulate the composition of ECM proteins and can also modify existing proteins, forming protein carbonyls (carbonyl groups) through exposure to cigarette-derived lipid peroxides and carbonyls [102]. The existence of protein carbonyls is linked to both the pathogenesis of COPD and macrophage retention and adhesion [113,114,115]. The latter reduces the ability to perform phagocytosis and promotes susceptibility to bacterial infections [113]. Efferocytosis is also impaired, reducing the clearance of apoptotic cells such as apoptotic neutrophils. The ineffective clearance of cell debris leads to necrosis and necroptosis [91,113,116,117]. This causes the release of toxic contents, further driving inflammation and the damage of lung tissue [113,118].

Taken together, these findings suggest that modifications (e.g., the addition of carbonyl groups) to the ECM disrupt the ability of macrophages to clear apoptotic neutrophils. This is a critical step, enabling tissues to return to homeostasis after inflammation. This causes secondary necrosis and necroptosis, releasing molecules that cause tissue damage and degrade the integrity of the lung, leading to ECM, tissue remodeling, and emphysema. This in turn disrupts the apoptotic surface cell markers and signals for phagocytotic clearance, causing apoptotic cells to undergo further necrosis and necroptosis, leading to emphysema [91,119]. This has the potential to create a loop in which the system is unable to return to homeostasis (see Figure 2). This is further exacerbated if the individual continues to be exposed to irritants, such as cigarette smoke, environmental hazards, or infections that further impact the ECM. The whole range of other factors described is involved in various parts of this process and defining the most important will be crucial in defining the most effective therapeutic targets. It should be noted that emphysema and impaired lung function are some of the greatest effects of CS, and that these effects, directly associated with CS, are not mediated by PTMs or biomolecules and instead are a direct result of exposure to cigarette smoke. As a result of this, they are beyond the scope of this review.

## 5. Variations in COPD Phenotype

There are several different phenotypes of COPD, with each presenting different symptoms, molecular profiles and, in some cases, different modes of disease pathogenesis. Currently, the only phenotypes that are not found through clinical symptoms are discovered via screening for α-1 anti-trypsin deficiency (AATD) and assessing levels of blood eosinophils [3]. The identification of these phenotypes alters treatment recommendations, with altered pharmacotherapy being used to improve patient quality of life (QOL) [3]. The use of treatment approaches based on disease phenotype is in its infancy, and more in-depth molecular characterization techniques will open new avenues for applying precision medicine to COPD and other respiratory diseases that share similar features, such as severe asthma [5,120].

### 5.1. AATD

AATD is a hereditary condition, with the key characteristic of reduced levels of AAT in circulation and lungs. AAT is a serine protease inhibitor that is made exclusively in the liver, that is released into the blood, and that is usually found in high concentrations in human sera [121,122]. Its key function is to inhibit proteolytic degradation through the targeting of NE, proteinase-3 (PR3), and cathepsin-G (CG). COPD is characterized by a protease–anti-protease imbalance. These three proteases are associated with tissue destruction and implicated in the pathogenesis of COPD, with NE being a key factor in the development of emphysema through the degradation of the ECM and elastase [93,123]. Other impacted pathways, important in COPD pathogenesis, include the inhibition of caspase-3, which inhibits the apoptosis of alveolar endothelial cells. This may be a key factor in the development of emphysema through reducing the death of structural lung cells [124,125]. The early diagnosis of AATD is linked to positive outcomes as it can reduce the likelihood that the individual begins smoking and encourages the avoidance of pollution [122]. Diagnosis also triggers treatment through augmentation therapy, using the intravenous infusion of AAT to provide a protective factor against further damage in the lung [122,126]. Without early intervention, it is not uncommon for patients with AATD to develop emphysema, especially if chronically exposed to cigarette smoke or pollutants.

### 5.2. Blood Eosinophils

Eosinophils are immune cells that have roles in inflammatory responses in 10–40% of patients with COPD [127,128,129]. Blood eosinophils are currently recommended as a precision medicine approach with which to inform clinicians of when to begin ICS treatment [3]. Although their precise roles in COPD are not fully understood, there is a clear link between patients with high levels of eosinophils and positive responses to ICSs [127]. The variability in eosinophil levels is a potential explanation of why patients with COPD have mixed responses to ICSs [128]. Blood eosinophils are currently recommended as a precision medicine approach with which to inform clinicians of when to begin treatment with ICSs [3]. In contrast to this, however, some clinicians argue that blood eosinophils are not reflective of lung tissue eosinophils and believe that there is no association between blood eosinophils and worse outcomes/treatable traits [130]. It should be noted that, although eosinophilic COPD has some similar characteristics to asthma, such as a sensitivity to ICSs, there are significant differences in the subtypes of eosinophil that circulate in patients with COPD compared to asthma patients, with the subtypes being inflammatory and resident eosinophils [131]. Furthermore, eosinophil levels are significantly higher in patients with asthma compared to COPD patients, specifically inflammatory eosinophils [131,132,133]. Recent studies employing the use of biologics have identified the inhibition of interleukin-4 and interleukin-13 as a pathway that can be targeted to reduce COPD symptoms in patients presenting a high eosinophil count [129,134]. It is unknown if this is a treatment specific to eosinophilic COPD due to the exclusion of a non-eosinophilic COPD cohort in the clinical trial [129,134].

AATD and eosinophilic COPD are currently the only ones clinically used to address treatable traits in COPD. Further research into the characterization of different phenotypes using biological pathways and lung physiology may lead to critical opportunities for differentially treating COPD. The presence of multiple different phenotypes highlights the fact that current drugs are insufficient and also being used incorrectly without phenotyping being performed. Numerous other phenotypes have been identified, where different approaches would result in more positive patient outcomes. One is the frequent exacerbator, where patients have more frequent exacerbations (>2 per year) than other patients. In this phenotype, there are four potential biological sub-phenotypes: bacterial, viral, eosinophilic and pauci-inflammatory [135]. Only one of these has an identified treatable trait that is screened for routinely (being eosinophilic). Further research on these sub-phenotypes to determine the different pathophysiology and distinct biomarkers would enable studies to target them specifically and enable large-scale studies of mixed phenotypes to define significant differences between internal cohorts. A wide range of other phenotypes have been identified and proposed, with distinctions also being made based on disease and structural differences, e.g., bronchitis versus emphysematous, and the type of remodeling seen in the airways, parenchyma, and pulmonary vessels/small vessel tree [136]. It is clear that COPD is a complex disease, with a high level of variation between patients. Thus, phenotypes need to be characterized in order to direct clinical trials and drug design towards individualized treatment.

## 6. Current Pharmacological Interventions

At present, COPD is an incurable disease. Current treatments aim to reduce the symptoms of the disease, slow progression, and reduce mortality, displaying varying levels of success. Despite this, randomized clinical trials for these treatments, attempting to demonstrates a reduction in mortality, have been unsuccessful in most cases. A notable exception to this was a clinical trial analyzing the impacts of fixed-dose triple-combination therapy using long-acting β2 agonists (LABAs), long-acting muscarinic agonists (LAMAs), and inhaled corticosteroids (ICSs), compared to a control group undergoing dual therapy of LABA with ICSs [137]. β2 agonists are an old class of drug with bronchodilator effects that have been employed since the early 1960s to treat asthma [138]. Similarly, ICSs have been the most effective treatment for asthma since their introduction in the 1970s [139]. Anti-muscarinic drugs were first used to treat respiratory diseases as early as the 17th century in Egypt, and were first used specifically for the treatment of asthma in Britain in 1802 [140,141]. While the initial implementation of combined therapy involving LABAs, LAMAs, and ICSs is related to the treatment of asthma, clinical trials to determine the efficacy of these drugs in treating COPD did not begin until the late 1990s to early 2000s, showing that the treatment of COPD is a relatively new field [142,143,144]. Emerging drug-based treatments for COPD include the usage of biologics such as dupilumab, with results indicating an increased quality of life and improved lung function when used alongside inhaled triple therapy [129,134]. Despite the positive initial outlook when using biologics to treat COPD, the heterogenicity of this method may reduce its efficacy in cohorts that do not present currently identified treatable traits. Thus, new and more effective drugs are urgently needed for COPD. An improved understanding of why these drugs have their effects could influence and assist in drug development for COPD. While there are a number of new drugs and emerging treatments currently in varying stages of clinical trials, this is beyond the scope of this review.

### 6.1. β2 Agonists

β2 agonists are bronchodilators that target β2 receptors on ASM cells. They are generally separated into the subgroups of Ultra-LABA (indacaterol, olodaterol, and vilanterol), LABA (arformoterol, formoterol, indacaterol, olodaterol, and salmeterol), and short-acting β-agonists (SABA, fenoterol, levalbuterol, salbutamol, and terbutaline) [3,145,146]. Generally, for patients with COPD, a LABA is often prescribed as a maintenance drug alongside other treatments, such as ICS, or/and with SABAs for use during exacerbations [3,147,148]. As COPD is not currently curable, the use of drugs with preventative modes of action (maintenance drugs) as part of daily care is critical for maintaining a good QoL. Unfortunately, many prescribed maintenance treatments are ineffective, and the pitfalls of their use when they do not reduce symptoms or change the disease state is that any adverse side effects may be compounded or worsened in the long term. Despite this, studies of β2 agonists, as a maintenance drug for COPD, indicate a reduction in symptoms and improvement in QoL [149]. Adverse effects from the use of β2 agonists include increased heart rate and palpitations, reflex tachycardia, and a mild drop in the partial pressure of oxygen (PaO_2_) [3,150]. While for most patients these risks are considered acceptable, in patients suffering from heart failure, some clinicians recommend using alternative treatments, such as anti-muscarinic drugs [151]. There are no official guidelines indicating that β2 agonists should be avoided in patients with heart failure, but several studies have called for further research on β2 agonists and any adverse effects on patients with heart failure [152,153].

### 6.2. Anti-Muscarinic Drugs

Anti-muscarinic drugs, much like β2 agonists, are bronchodilators used to relieve and prevent bronchoconstriction [154]. They are separated into SAMAs (ipratropium bromide, oxitropium bromide) and LAMAs (tiotropium bromide, aclidinium bromide, glycopyrronium bromide and umeclidinium bromide), with SAMAs being used like SABAs and only administered as needed. Like LABAs, LAMAs are common maintenance drugs and are often used with LABAs in combination therapy, which has been found to improve lung function compared to monotherapy [155]. Anti-muscarinic drugs largely have a positive safety profile and side effects are often disregarded by GOLD (2023) guidelines, with main ones being a dry mouth and potential urinary issues such as urine retention [156].

### 6.3. ICSs

ICSs are often combined with LABAs or LAMAs to reduce exacerbations in COPD. Dual therapy attempts to harness the benefits of the anti-inflammatory effects of ICSs and the bronchodilator effects of LABAs and LAMAs [157,158,159]. The current consensus is that using ICSs alone does not have any impact on the progression or mortality of COPD and that other drugs such as β2 agonists are required to induce a positive effect [160,161]. The current GOLD guidelines (2023) indicate that the overall benefit of ICSs is unclear due to a range of strong adverse effects, such as increases in pneumonia incidence, oral candidiasis incidence, and a significant increase in factures of the hip and upper extremities [160,162]. ICS exposure is a significant risk factor for the development of osteoporosis and the increase in fractures is a significant issue for the elderly [163,164]. This is due to the impact of ICSs on growth and sex hormones, which has to a cascading effect, reducing bone formation and increasing bone resorption [163,164]. With short-term treatment, bone loss is minimal; however, COPD requires regular long-term treatment [163,164]. Another significant side effect associated with the extended use of ICSs is an increase in pneumonia. While smoking and COPD increase the risk of developing pneumonia, ICS use further increases this risk [161]. The current consensus is that these adverse effects are outweighed by the reduction in exacerbations frequency and the consequences in patients who respond positively to ICS, e.g., eosinophilic COPD patients [165].

### 6.4. Biologics and Anti-IL-4/13R Antibodies

Until recently there has been no conclusive evidence that biologic therapies in COPD improve patient outcomes [134]. Recently, a clinical trial testing dupilumab, a monoclonal antibody targeting interleukin 13(IL-13) and interleukin 4 (IL-4) pathways, was able to reduce type-2 inflammation and exacerbations and increase lung function in COPD patients with a high eosinophil count when used alongside inhaled triple therapy [129,134]. Further research on biologics such as dupilumab needs to be conducted on a varied cohort to determine the efficacy of the drug in patients with a variety of COPD traits, e.g., a lack of a high eosinophil count and different ages. In a similar trend, other biologics such as benralizumab and mepolizumab have shown in clinical trials that they are able to reduce the occurrence of COPD exacerbations in patients with high blood eosinophil counts (≥300 cells/μL for benralizumab and ≥150 cells/µL at screening or ≥300 cells/µL in the a year for mepolizumab) [166,167].

## 7. The Need for New Drugs

While the above drugs provide a foundation for ongoing treatment, they are targeted at managing disease symptoms rather than curing the disease or stopping its progression. This is because the mechanism of COPD pathogenesis is not fully understood. Thus, there is a clear gap in terms of the knowledge that is needed to develop and test drugs that can reverse the chronic features of COPD or stop progression from early into later stages. With the mechanisms of pathogenesis incompletely understood, there is a strong need to characterize the biology and molecular physiology of progression from a healthy to a COPD state. Hypothesis-driven research and the interrogation of mouse models that most accurately reflect human smoking and COPD have led to the identification of promising new therapeutic targets and the development and testing of potential new therapies. These can be validated and translated in human cells and tissues in vitro and ex vivo. Furthermore, multiple different omics-based approaches are being utilized to characterize the molecular changes that occur during COPD initiation and progression on a more global and holistic scale. There is a critical need to continue this work to develop, validate, and translate more effective treatments and perform more in-depth biological and molecular characterization to discover druggable targets.

### 7.1. Animal Models to Define Therapeutic Targets and Develop and Test New Therapies

Nose-only models of cigarette smoke exposure are the most accurate models that mimic the inhaled route of exposure [1,2,45,65,66,68,69,70,74,75,76,79,91,104,168]. They see the development of the chronic features of COPD in shorter timeframes (8–12 weeks), reducing the length of exposure and making development, progression, exacerbation and treatment studies feasible [45,54,56,65,69]. If mice are exposed to cigarette smoke for 8 weeks, they develop all of the chronic features of COPD. If they are exposed for another 4 weeks the disease becomes more severe. If they are rested for 4 weeks, the features remain constant. This gives two therapeutic windows of opportunity for the testing of therapies to stop progression or reverse the disease. Mice have the same gene signatures as human COPD patients. They responses only have stress over the first five days, and do not display nicotine withdrawal symptoms. Whole-body exposures situate mice in a smoking box; they are not isolated or restrained and their whole bodies are exposed [119,169,170]. This deposits nicotine and particulate matter onto the mouse fur, which they groom and ingest. There is also a build-up of anti-inflammatory carbon monoxide. It takes four to six months to develop the chronic features of COPD in these models, precluding many long-term exacerbation and treatment experiments, or the development of other smoking-related disease such as lung cancer and pulmonary fibrosis [171]. It takes an inordinate amount of time, funding, and effort, as well as a large number of mice, to establish these models and they should be used to their full capacity wherever they are established. They are valuable for assessing the pathogenesis of disease in the lung, but also the influx of inflammatory cells from the bone marrow, blood, spleen and thymus, and the contributions of other organs. They have complete immune and vascular systems that can be used to assess interactions with other smoking-related diseases such as lung cancer, Crohn’s disease in the gut, autoimmunity (amongst many other issues), and the impact of smoke (or other exposures) in pregnancy on the offspring at different ages [171,172,173]. Genetically modified mice can be used to definitively demonstrate the role of and potential for therapeutic targeting or the manipulation of specific factors in COPD (e.g., AAT). They are not biased by what we already know, and develop the critical features of remodeling, emphysema, and impaired lung function. It should, however, be noted that there are some key anatomical differences when comparing mice to humans, such as the absence of bronchioles and fewer submucosal glands [174].

### 7.2. Human Ex Vivo and In Vitro Studies

We can examine primary tissues and cells from patients in bronchial or nasal biopsies or brushings, broncho-alveolar lavage, sputum, exhaled breath condensate, blood/serum/plasma, urine, faeces, or explanted tissues from lung transplant patients [175,176,177,178,179,180,181,182]. These are crucial for examining changes that specifically occur in patients. Human clinical trials are critical for the translation of new therapies into the clinic [177]. They are logistically challenging and expensive; it is extremely difficult to perform longitudinal studies, and they are impacted by patient heterogeneity. Results from nasal and blood samples often do not replicate events in the lung. Explanted lungs are taken from patients with severe GOLD 3 and 4 disease or with lung cancer, and they are very valuable [67,91,119,183]. However, this is not suitable for studying disease development, progression, or exacerbation, and may be confounded by cancer-related pathways. In this context, lungs from unused transplants can serve as a valuable resource.

Precision-cut lung slices can be used from humans and mice (+/− COPD) and kept alive in a lab for the assessment of specific factors or treatments [184]. One of the limitations of using isolated lung slices is that they lack a functioning immune system or input from other organs. These methods also include in vitro cultures of isolated cell types. The high-throughput or gold-standard submerged cultures are 6-week air–liquid interface (ALI) cultures, where primary respiratory cells are cultured from basal cells and differentiated in epithelia [54,55,56,57,60,77,78,185,186]. Lung-on-a-chip set-ups/organoids are similar to ALI cultures but add a mechanical physiological function, where liquid growth media is pumped past cells to resemble the movement of blood or expanded/contracted when grown on a membrane that can be inflated [187,188]. This technology is in its infancy, with most studies still focusing on method development and the most promising advanced models only using 2 or a maximum of 3 cell types. All these methods are used in primary discovery studies and are harnessed to validate and translate findings from mouse models or human primary tissues from in vitro/ex vivo models. All are influenced by the heterogeneity of donors, and co-cultures with isolated immune cells can be achieved but suffer from further increased variability. This requires many cultures to be used, which is difficult. They do not have the functional immunity of vascular systems or the microenvironments present in vivo. They cannot be used for high-throughput studies and answer limited questions about the interactions of the lung and the immune system, offering no information about other organs. Results are biased by the cell types included and simply defined media that do not represent human tissue fluids. The contrived interactions between cells may be useful, but are unnatural. These systems cannot be used to unravel the complex processes that lead to remodeling, emphysema, and lung function changes.

These in vivo mouse models (especially nose-only variants) should be used in combination with primary human and ex vivo/in vitro studies to define events in vivo and validate and translate them into human tissues and cells ex vivo and in vitro. This remains the optimal process with which to better understand COPD pathogenesis and develop and test new therapies [54,56,58,64,65,67,70,74,76,91,119,183,189,190,191].

## 8. Current Status of Characterizing the Molecular and Cellular Landscape in COPD

### 8.1. Biomarker Characterization

Biomarkers are molecules that occur within the body that can be used to make clinical recommendations or diagnoses. Generally, biomarkers are assessed through pathological testing via blood, urine, and sputum sampling or biopsies [192,193]. They have the potential to be part of routine examinations for “at risk” patients who are yet to present symptoms, such as those proposed for Alzheimer’s disease [194]. While it may seem like biomarker testing should be routine for everyone, there is a significant financial cost of implementing mass testing. This would be compounded by the substantial portion of countries not offering medical support for diagnostics and treatment, such as the USA [195,196]. Thus, biomarker testing may need to be targeted. Traditional hypothesis-driven research has investigated the biomarkers of COPD development and progression. Recently, eosinophil levels have been measured to determine ICS and biologic sensitivity, or AAT for AATD [122,128]. These biomarkers, however, are not relevant for significant numbers of patients with COPD and are more suitable for determining their phenotype [197]. One biomarker that shows strong potential in COPD assessment is the soluble receptor for advanced glycation end products (sRAGE). It appears to be a robust biomarker associated with emphysema and airflow obstruction, and large-scale studies incorporating a variety of ethnicities are needed to determine its biomarker value and role in COPD [198].

New multi-omics techniques are being used to more globally and holistically map pathogenesis through entire disease pathways and networks [199]. They are being widely adopted to discover biomarkers and druggable targets and include bulk single-cell and spatial proteomics, transcriptomics, metabolomics and lipidomics. Currently, omics-based biomarkers are largely in pre-clinical to clinical trial stages due to difficulties in validation across cohorts. This is in part due to inherent differences between patients, e.g., age, gender, race, comorbidities, disease stage [197], and COPD heterogeneity [200]. This is a common theme in biomarker identification across diseases, such as multiple sclerosis [201]. As different phenotypes likely have different mechanisms of pathogenesis and disease progression, it is unsurprising that patients present different proteomic, lipidomic, transcriptomic, and metabolomic profiles. While this makes diagnosis with biomarkers difficult, it highlights the need for a multiple-biomarker approach when diagnosing COPD [202]. Phenotype-specific biomarkers may have a critical advantage in treating the disease early. However, precision medicine approaches, which tailor pharmacological recommendations for the phenotype patients are presenting, offer a significant advantage in terms of drug efficacy and patient response [203,204].

### 8.2. Potential Therapeutic Targets

In-depth molecular characterization has the potential to reveal novel therapeutic targets for further research and testing. Current molecular characterization methods have highlighted key molecular pathways such as those influencing lung surfactants, and immune cells such as macrophages and neutrophils, as key to the pathogenesis and development of COPD.

### 8.3. Lung Surfactants

Lung surfactants are complex mixtures of proteins and lipids that function as thin interfaces lining the alveolar walls to modulate critical lung properties, such as surface tension and air–liquid gas exchange [205,206]. While alveolar wall composition plays a critical role in the pathogenesis of COPD, surfactants also provide an avenue for disease pathogenesis [205]. Molecular measurements of lung surfactants show that their proteins and lipids change in concentration in patients with COPD compared to healthy controls [207,208]. One study used immunoblot and ELISA to find that pulmonary surfactant protein-A (SP-A) and -D (SP-D) increase in the blood serum as COPD GOLD stages progress but decrease in the BALF. These changes directly correlate with increased concentrations of C-reactive protein (CRP), a common biomarker of general inflammation and infection [209]. This is highly clinically relevant, as these two proteins are associated with the innate immunity of alveoli [210]. Other studies reported conflicting results, whereby serum SP-D levels were not associated with the development of COPD, but increased levels may induce a slower decline in FEV1 in patients with COPD [211]. The conflicting results indicate that more research is needed to define their suitability as biomarkers or therapeutic targets for COPD.

### 8.4. Alveolar Macrophages

The transcriptomic and lipidomic profiling of COPD alveolar macrophages (AMs) has identified significant differences between COPD GOLD grades [212]. Between GOLD grades 1–4, there is a phenotypic shift in AMs that is distinct for each grade. AMs are critical for maintaining lung homeostasis, with roles in inflammation and the catabolism of lung surfactants [212,213]. Since lung surfactants are critical for maintaining optimal lung function, the breakdown of the proteins and lipids critical to their composition could greatly disrupt lung function. Lipidome composition changes can cause a shift in the functional responses of cells with Ams, increasing lung surfactant catabolism and therefore lung surface tension and reducing protection against pathogens [210,212,214]. Another study found that there was no significant change in the numbers of AMs present in the alveolar region, and the role AMs have in the progression of COPD was only determined using transcriptomics and lipidomics [212]. With no direct change in abundance, this suggests that that the phenotypic shift between grades may play a role in pathogenesis and the progression through GOLD grades 1–4.

### 8.5. Macrophage Phenotypes

Macrophages have a range of phenotypes that heavily influence their properties and responses to stimuli [215,216]. In a healthy individual, alveolar macrophages do not exhibit features of an M1 or M2 phenotype and can be referred to as non-polarized or M0; they, therefore, have the capacity to change phenotypes in response to signaling mechanisms [215]. Macrophages are best referred to by their specific features, rather than as M1 or M2 (e.g., Figure 3). M1 and M2 markers can co-occur on AMs from patients [215], which may indicate a dysfunctional or damaged macrophage or potentially a phenotype shift specific to COPD.

### 8.6. M1-like Macrophage Phenotypes

Those in the M1-like phenotype are referred to as “classically activated” macrophages and are closely associated with pro-inflammatory and cytotoxic responses due to their ability to secrete pro-inflammatory cytokines, chemokines, and cytotoxic molecules such as tumor necrosis factor, interleukin-1 β, macrophage inflammatory protein-2, inducible nitric oxide synthase, cyclooxygenase-2, and reactive oxygen species [217,218]. The release of these molecules into alveolar spaces is associated with inflammation, increased anti-microbial activity, and lung damage [218]. This appears to be a damaging phenotype, but in the healthy state it is only expressed to fight infections acutely, with damage being repaired by M2 macrophages that increase in abundance to combat the self-damage [217]. M1 phenotype expression increases in response to cigarette smoke exposure, and decreases upon the cessation of smoking in non-COPD or mild COPD patients [215]. This presents an avenue for further research, with M1 polarization in AMs serving as a potentially druggable target for reducing inflammation in COPD.

### 8.7. M2-like Macrophage Phenotypes

In contrast, the M2-like phenotype contains “alternatively activated” macrophages and they are associated with anti-inflammatory effects, phagocytosis, and tissue repair/remodeling. They may be categorized into five different subgroups with different effects: M2a, M2b, M2c, M2d, and M2f (Figure 3) [219]. The proteomic analysis of M0, M1 and M2 subtypes indicates that they have their own protein expression profiles with the M2 subtypes, being similar to each other [220].

### 8.8. M Unknown

A study on human BALF from COPD patients found a significant increase in a macrophage type that did not present any known M1 or M2 macrophage signatures and had a distinct transcriptomic signature [216]. This provides further evidence of macrophage phenotypes specific to COPD that need further phenotypic characterization in order to better understand their role in pathogenesis. It is suggested that this uncharacterized phenotypic shift may be a key factor in either the manifestation of disease symptoms or the pathogenesis of COPD [216].

### 8.9. The Need for More Detailed Macrophage Characterization

The presence of unknown macrophage phenotypes in COPD highlights the clear need for an improved characterization and understanding of macrophage phenotypes beyond these classifications. One way to study phenotypes is to look at post-translational modifications (PTMs) associated with distinct functions or different disease states. This can be achieved using the most representative animal models and cells from human COPD patients and by using hypothesis-driven and state-of-the-art multi-omics technologies [199]. In addition, PTMs of proteins are largely unexplored in COPD but influence the function and phenotype of macrophages, indicating that they have a more complex phenotype than is currently understood [221]. As the same molecule can exist in a system with different PTMs inducing different functions, PTMs present a way to more comprehensively assess cellular changes induced in diseased states [222].

### 8.10. Macrophage Subpopulations in the Lung

Supporting the hypothesis that phenotype shifts play a critical role in COPD pathogenesis, we discovered that macrophage expression changes significantly between the small airways and bronchioalveolar lavage (BAL) lumen of COPD patients [178]. Our study found that the small airway of smokers with normal lung function and smokers with COPD underwent a phenotypic shift to a majority M1 macrophage population; this is in contrast to the control, where the majority showed M0 macrophages [178]. The BAL lumen presented a different population, with normal lung function smokers and COPD smokers presenting a majority M2 profile [178].

### 8.11. Neutrophils

Neutrophils have been identified as potentially druggable targets for COPD. Neutrophils in the blood and BALF at early stages of COPD present phenotypic changes that influence degranulation, RHO GTPase signaling, and translation [64]. Both phenotypic shifts and increased neutrophil abundance have been shown to correlate with a decrease in lung function [64]. While a large portion of this has been performed in murine models, the assessed changes in human blood and BALF neutrophil transcription present an overlap with murine models of COPD [64]. Neutrophil phenotyping identified an association between three neutrophil states (N2S/ISG, N4S/G0S2, and N5S/S100A12) that are directly associated with COPD symptom manifestations not found in more acute respiratory conditions [64].

This defines immune cells such as AMs and neutrophils as potential druggable targets for COPD [64,212]. Further characterization of the changes to immune cells such as AMs and neutrophils will produce a more complete understanding of how cellular changes impact inflammation, and thus the progression and pathogenesis of COPD.

## 9. Novel Therapeutic Targets

The recent interrogation of mouse models of COPD and complementary human studies has defined promising new therapeutic targets which, as stated previously, are urgently needed.

These include cell-derived responses from mast cells and their proteases; T, B, NK and innate lymphoid cells; molecular-based responses involving inflammasomes; oxidative stress, IL-22, TRAIL, micro(mi)RNAs and the pathways they control; stress granules; and amine and lysyl oxidases. Finally, chronic inflammation is also influenced by the lung and gut microbiomes and the gut–lung axis.

### 9.1. Mast Cells

A range of features associated with mast cells have been indicated to play roles in COPD. Mast cell proteases such as tryptase (Prss31) and chymase (Hcma)1 have been shown to influence COPD’s severity and pathogenesis by exerting a pro-inflammatory effect and instigating the expression and release of tumor necrosis factor α from lung macrophages [45,66,67]. We have also recently demonstrated the critical role of increased Toll-like receptor 7 (TLR7) responses in COPD, driving the destruction of alveolar epithelial cells that promote pulmonary emphysema [65].

TLR7 is a single-stranded RNA receptor that plays a role in generating immunity against single-stranded RNA viruses. Research indicated that abundance increases the number of TLR7-deficient mice presenting with a decreased severity of COPD and emphysema in a CS exposure model. In contrast, the inhalation of imiquimod, an agonist for TLR7, caused mice to present with emphysema, even in the absence of CS exposure [65]. This model of inducing emphysema with imiquimod is shown to be reduced in effectiveness with mice that are deficient in mast cell protease-6 or when wild-type mice undergo treatment with cromolyn, a mast cell stabilizer [65]. Additionally, mice treated with an anti-TLR7 monoclonal antibody present with a reduction in CS-induced emphysema and experimental COPD and display a reduction in the accumulation of pulmonary mast cells [65]. Existing datasets from COPD patients present an increase in TLR7 mRNA, alongside an increase in TLR7+ mast cells in COPD diseased lungs, directly associated with increased severity of COPD [65]. Similarly, Silgec-8 is an inhibitor on mast cells that has been indicated to be associated with the activation of mast cells and successive immune cell recruitment, airway inflammation, and the onset of pulmonary fibrosis [68].

### 9.2. NK Cells

As macrophages, NK cells are a vital component of the innate immune system, particularly within the innate lymphoid cell (ILC) group and are known for their ability to respond rapidly to a wide range of challenges, including tumor cells and viral infections [223]. NK cells identify and eliminate cells that are stressed, transformed, or infected, primarily through the release of cytotoxic granules and the production of inflammatory cytokines, in particular the large-scale production of cytokines that can prime other immune cells such as IFN-γ and GM-CSF, which can contribute to exacerbated inflammation [224].

In the context of cancer, NK cells can recognize and destroy malignant cells without prior sensitization, making them a focus in cancer immunotherapy research. Similarly, their role in combating viral infections involves the detection and killing of virus-infected cells, contributing to the containment and resolution of viral diseases and other infections. In COPD, the anti-viral capacity of NK cells was recently observed to be dysfunctional [58]. However, in COPD, there is often a state of chronic inflammation and tissue damage, where NK cells might contribute to disease exacerbation through their pro-inflammatory actions [225]. Dysregulated NK cell activity in COPD can lead to tissue damage and exacerbate the inflammatory state, potentially worsening the disease. Given their dual role, NK cells present a complex but promising target for therapeutic interventions in COPD. Modulating NK cell activity to enhance their protective roles, while minimizing their contribution to chronic inflammation, could open new avenues in the treatment and management of COPD and related chronic inflammatory conditions.

### 9.3. T, B, and Innate Lymphoid Cells

Both adaptive and innate immune responses have been indicated to play roles in the pathogenesis of COPD [69]. Specifically, our group investigated the impact that B lymphocytes, T lymphocytes, and group 2 innate lymphoid cells have regarding airway inflammation, airway remodeling, and changes in lung function [69]. Experimental data indicate that these specific immune pathways increase airway collagen deposition/fibrosis, but do not influence airway inflammation [69].

### 9.4. Inflammasomes

Inflammasomes are innate immune system molecules that consist of a sensor protein, adaptor, and a pro-caspase-1. When activated, they exert an inflammatory effect on the body [176,226,227]. In a functional immune system, effectors initiate inflammation to protect from pathogens or in response to inflammatory signaling pathways induced by cell death [168,226,227]. Excessive inflammasome signaling has been indicated to play a major role in a range of disease such as cancer, cardiovascular diseases, neurodegenerative diseases, and COPD [168,226,227].

Murine models of COPD have highlighted several inflammasomes that influence the pathogenesis and/or severity of COPD. Key inflammasomes that present druggable pathways include the Leucine-rich repeat protein 3 inflammasome, the pyrin domain containing protein 3 inflammasome, and the Aim2 inflammasome [47,168,176,228]. Drugs such as Daphnetin and Schisandrin A have been indicated to provide an anti-inflammatory effect in murine models by targeting the NOD-like receptor protein 3 inflammasome and they present as potential drug candidates for COPD [229,230].

### 9.5. Oxidative Stress

Oxidative stress is a key factor that has been implicated in the pathogenesis of both cigarette smoking-based and environmental pollutant-based COPD [72]. Oxidative stress in COPD occurs due to an increase in both endogenous and exogenous reactive oxygen species (ROS). Exogenous ROS in COPD are caused by external factors such as cigarette smoke inhalation and other environmental factors such as air pollution from exhaust fumes [71,72,231]. In endogenous ROS, the increase is caused by damaged/dysfunctional mitochondria or in some cases, such as severe asthma, by immune cells [72,73]. Treatment approaches that modulate oxidative stress through the use of antioxidants have been put forward as potential treatments for COPD [71,72,73].

### 9.6. IL-22-TRAIL (Cytokines)

Interleukin (IL)-22 and the tumor necrosis factor-related apoptosis-inducing ligand (TRAIL) are both cytokines involved in COPD pathogenesis [74,75]. IL-22 has been clinically shown in COPD murine models to increase mRNA expression and protein expression in response to chronic CS exposure, with IL-22-deficient models presenting improved lung function and no airway remodeling when compared to the IL-22 mice [74]. TRAIL has been indicated to play a role in both airway inflammation and apoptosis; murine models of COPD indicate that TRAIL-deficient mice suffer from reduced pulmonary inflammation and airway remodeling when compared to TRAIL models [75].

### 9.7. Micro(mi)RNAs

Various research findings suggest that micro(mi)RNAs are involved in the pathogenesis of COPD [76,77]. Research identified that miR-21 is the second most up-regulated miRNA in the lungs of mice in a CS-induced model of COPD, especially in the airway epithelium and lung macrophages. The expression of miR-21 has also been shown to correlate with reduced lung function in human lung tissue taken from COPD patients. The preventative and therapeutic administration of a miR-21 inhibitor known as Ant-21 has been shown to result in reduced CS-induced miR-21 expression in mice, along with the suppression of airway immune cells (macrophages, neutrophils and lymphocytes) and the improvement of lung function in mice COPD models [76,77]. miR-328 has been shown to play a role in bacterial clearance, with targeted inhibition enhancing host defenses against microbial infection [70]. This presents an avenue to treat COPD patients suffering from frequent exacerbations related to microbial infections, effectively improving lung function [70].

### 9.8. Stress Granules

Primary bronchial epithelial cells in COPD patients present a reduced expression of protein kinase R (PKR), which leads to a decrease in PKR-mediated stress granules [78]. These stress granules play a critical role in modulating the induction of interferons such as IFN-β [78]. This pathway has been indicated to be a pathway for reducing exacerbations in COPD, reducing susceptibility to influenza infections if restored to its correct function through treatment [78].

### 9.9. Amine/Lysyl Oxidases

The inhibition of oxidases such as semicarbazide-sensitive amineoxidase (SSAO) and lysyl oxidase like 2(LOXL2) has been put forward as important for reducing COPD symptoms [79,80]. SSAO has been shown to reduce airway inflammation and lung fibrosis in mice, resulting in improved lung function [79]. In a similar manner, the inhibition of LOXL2 presents reduced collagen cross linking, effectively reducing lung fibrosis [80].

### 9.10. Lung and Gut Microbiomes/Gut–Lung Axis

The lung microbiome and the gut microbiome are two microbiomes that present targets for therapeutic intervention [81,82,83,84,85,86]. The lung microbiome in a healthy individual is believed to be transient, with regular immune clearance and re-seeding from the external environment [81]. In contrast to this, individuals suffering from respiratory diseases have a more persistent lung microbiome, presenting evidence that the immune system is not functioning correctly [81,83]. Specifically, the enrichment of the lung microbiome with taxa from the oral cavity is associated with inflammation [81,83]. The gut microbiome has been presented as a potential cause for this, with diet and other gut microbiome-alterative factors such as antibiotics presenting modulating effects on the immune system, including systemic and airway inflammation [83,84,86,87]. Specifically, an increased abundance of *streptococcus* has been shown to be associated with smoking, an increase in the abundance of *Lachnospiraceae* family is associated with reduced lung function, and a specific subset of the *Dorea* species is associated with an increase in inflammatory cytokine release [83].

### 9.11. Post-Translational Modifications in COPD

Post-translational modifications (PTMs) occur to a protein after it has undergone ribosomal synthesis. There are two major categories of PTM, namely, irreversible proteolytic cleavage and reversible covalent modifications [232,233], with the end result termed a “proteoform” to define it as distinctly different from the “unmodified” proteoform. Different proteoforms of the same protein have different functions. In COPD, several PTM-containing proteoforms are observed to increase in abundance, including those that undergo methylation, phosphorylation, and sumoylation [91,234,235,236,237,238]. There are a wide range of other pathologies where proteoforms are being investigated for their therapeutic targeting potential, such as Parkinson’s disease, cancer, and kidney disease [239,240]. For a more comprehensive discussion of proteoforms, see [241].

Proteolytic cleavage occurs through the activity of proteases, like endopeptidases and exopeptidases [242], and is critical for cellular activity. The key processes that proteases influence include enzyme activation/deactivation, cellular signaling, and inflammatory and immune responses [243]. This is because proteins are commonly synthesized as inactive precursors and cleaved to take an active form, or synthesized in an active form and cleaved to be inactivated [244]. Much like proteolytic cleavage, covalent modification has a critical role in regulatory mechanisms and cellular processes, with over 400 different PTMs discovered to date [233,245]. The dysfunction of either type of PTM is linked to the pathogenesis of a wide range of diseases, including chronic inflammation, cancer, cardiovascular disease, and musculoskeletal and neurodegenerative disorders [246,247,248].

PTM detection and characterization is an emerging field. The identification of those in relation to COPD is under-researched and there is an urgent need for better characterization of molecular and cellular pathways in order to guide therapy [249]. So far, there have been nine studies, all within the last five years (Table 2), that have assessed the role of PTMs in COPD. Few have used mass spectrometry to confirm this, and none have performed any untargeted PTM discovery. This is an issue, as targeted PTM analyses rely on some form of enrichment method, such as affinity capturing, to concentrate the PTMs to be analyzed through LC-MS/MS. This has the potential to produce selective results as the analysis is limited by the binding affinity of the reagents used [250]. Nevertheless, these issues are not unique to PTM analysis and occur in other analytical procedures such as immunoassays [251,252]. The potential for bias does not invalidate them as analytical approaches for molecular characterization, but does highlight the need for PTM discovery/analysis that does not suffer such biases. One avenue that shows significant promise for PTM research that removes binding affinity-induced bias is untargeted PTM characterization. Untargeted approaches are critical to identifying novel PTMs and are an essential tool with which to complete the characterization of COPD [253].

In other pathologies, such as cancer, PTMs are being explored to discover druggable targets, with the targeting of protein methylation and ubiquitinoylation showing significant potential for interrupting tumorigenesis and cancer development [254,255]. Additionally, PTMs such as sumoylation present a druggable target, with clinical trials for its inhibition underway to determine side effects and cancer response [256]. Trials like this are the result of the in-depth characterization of PTMs targeted as treatments for cancer [257]. With PTMs showing relevance in a wide range of cellular events, it stands to reason that this would also apply to COPD, with PTM characterization highlighting potentially druggable targets for further research. The usage of omics-based research using mass spectrometry to understand biological systems has been well documented, and readers can find further information in a multitude of in-depth reviews [258,259,260,261,262].


proteomes-12-00023-t002_Table 2Table 2Existing COPD PTM studies. These are from the last 5 years from a PubMed search of chronic obstructive pulmonary disease and post-translational modification.Year PublishedTitleMass Spectrometry?2020Cigarette smoke extract stimulates bronchial epithelial cells to undergo a sumoylation turnover [237]Yes2019Endoplasmic reticulum stress and unfolded protein response in diaphragm muscle dysfunction of patients with stable chronic obstructive pulmonary disease [263]No2022K63 Ubiquitination of P21 can facilitate Pellino-1 in the context of chronic obstructive pulmonary disease and lung cellular senescence [264]No2021LSD1-S112A exacerbates the pathogenesis of CSE/LPS-induced chronic obstructive pulmonary disease in mice [265]No2019Effects of concurrent exercise training on muscle dysfunction and systemic oxidative stress in older people with COPD [266]No2019Vitamin D protects against particles-caused lung injury through induction of autophagy in an Nrf2-dependent manner [267]No2019Carbocisteine improves histone deacetylase-2 deacetylation activity by regulating sumoylation of histone deacetylase-2 in human tracheobronchial epithelial cells [268]No2020Airway resistance caused by sphingomyelin synthase-2 insufficiency in response to cigarette smoke [269]No2020The protective effects of HBO1 on cigarette smoke extract-induced apoptosis in airway epithelial cells [270]No


## 10. Conclusions

COPD is a heterogenous disease that contains multiple different phenotypes with clinical implications. Current medication does not have the capability to stop or reverse COPD, which indicates a strong need for the development of new therapeutics. For new druggable targets to present themselves, a significant amount of work needs to be performed to characterize changes at a molecular level. Due to the complex nature of the disease and the need to study the onset and progression of COPD, lungs from late-stage COPD patients, in vitro cells, and lung-on-a-chip/organelle models are not suitable for determining personalized medicine approaches. This is because such a model that can mimic human smoking habits paired with a complex immune system is required. The nose-only mouse smoke exposure model presents the most suitable option as it reduces animal suffering through the significantly shorter time required to induce COPD symptoms when compared to the other alternative of whole-body exposure, which takes approximately twice as long as nose-only exposure and leads to-off target effects from the deposition of harmful particulate matter on the mice. As such, it presents the most promising model for determining biological changes that would translate into human studies, such as post-translational modification characterization, and a range of other novel therapeutic approaches, such as the use of mast cells and their proteases; T, B, NK and innate lymphoid cells; macrophages and neutrophils; inflammasomes, oxidative stress, IL-22, TRAIL, and micro(mi)RNAs and the pathways they control; stress granules; amine and lysyl oxidases, lung and gut microbiomes; and the gut–lung axis. Ultimately, to effectively treat a heterogenous disease such as COPD, a more in-depth characterization of molecular changes occurring during its pathogenesis is required. This would allow druggable targets to be identified, opening new avenues for precision medicine that has the potential to improve patient outcomes.

## Figures and Tables

**Figure 1 proteomes-12-00023-f001:**
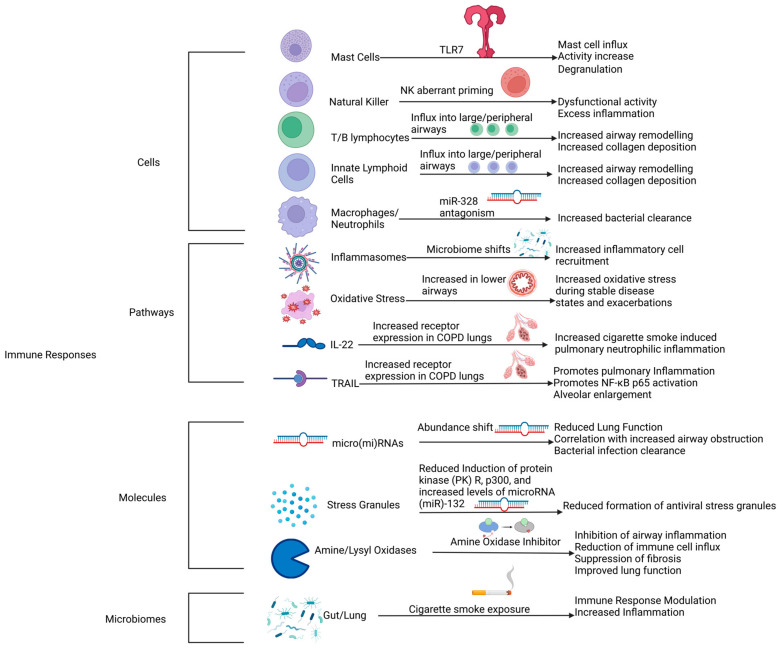
A visual representation of the research performed at Centre for Inflammation, Centenary Institute and University of Technology Sydney, Faculty of Science and the relevance to COPD. This was created with BioRender.com.

**Figure 2 proteomes-12-00023-f002:**
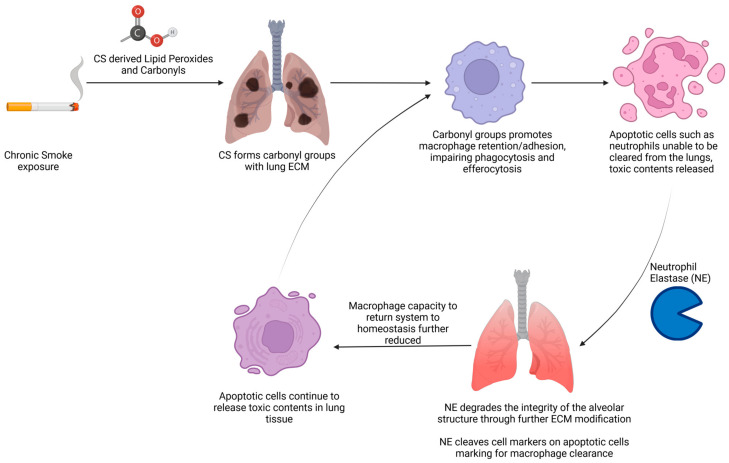
A visual representation of the cascading impact carbonyl groups exerts over lung immune function, driving inflammation and forming a disease state. This was created with BioRender.com.

**Figure 3 proteomes-12-00023-f003:**
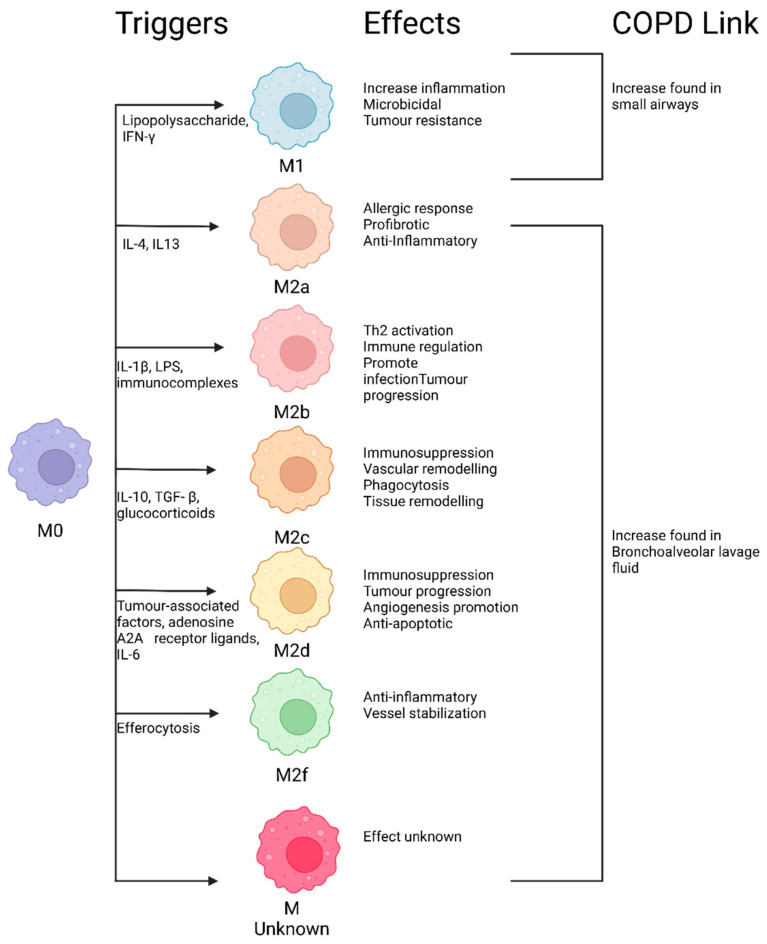
Macrophage differentiation pathways from M0 to M1 and M2 subtypes, with associated triggers for the phenotype shift. This was created with BioRender.com.

**Table 1 proteomes-12-00023-t001:** Symptoms of most common respiratory diseases.

	Inflammation	Airway Remodeling	Cough	Dyspnoea
COPD	✓ [3,10,11]	✓ [3,10]	✓ [3,10]	✓ [3]
Asthma	✓ [12,13,14,15]	✓ [12,13,14,15]	✓ [14]	✓ [14]
IPF	✓ [16,17,18]	✓ [16,17]	✓ [17,18]	✓ [17,18]
CF	✓ [19,20]	✓ [19,20]	✓ [21]	✓ [22]

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
