# Peer review of "The Current Molecular and Cellular Landscape of Chronic Obstructive Pulmonary Disease (COPD): A Review of Therapies and Efforts towards Personalized Treatment"

_proteomes, 2024, doi:10.3390/proteomes12030023_

Round 1

Reviewer 1 Report

Comments and Suggestions for Authors

The manuscript provides a comprehensive review of Chronic Obstructive Pulmonary Disease (COPD), detailing its molecular landscape, current diagnostic methods, therapies, and the movement towards personalized treatment strategies. The manuscript is well-organized, and authors effectively outline the complexity of COPD, its diverse phenotypes, and the role of omics-driven methodologies in enhancing our understanding and treatment of this condition.

I have minor suggestions:

1.  While the manuscript thoroughly reviews current therapies, it could benefit from a more a brief discussion on emerging treatments and their clinical trial statuses.

2. The inclusion of more figures and tables, particularly those illustrating molecular pathways, diagnostic techniques, and therapeutic mechanisms, would enhance the reader's understanding.

Comments on the Quality of English Language

English language is fine. Minor typographical error could be corrected. 

Author Response

Reviewer 1

The manuscript provides a comprehensive review of Chronic Obstructive Pulmonary Disease (COPD), detailing its molecular landscape, current diagnostic methods, therapies, and the movement towards personalized treatment strategies. The manuscript is well-organized, and authors effectively outline the complexity of COPD, its diverse phenotypes, and the role of omics-driven methodologies in enhancing our understanding and treatment of this condition.

I have minor suggestions:

  1. While the manuscript thoroughly reviews current therapies, it could benefit from a more a brief discussion on emerging treatments and their clinical trial statuses.

We thank the reviewer for this suggestion, as the focus is not on emerging treatments a sentence has been added to state this (line 323-324)

  1. The inclusion of more figures and tables, particularly those illustrating molecular pathways, diagnostic techniques, and therapeutic mechanisms, would enhance the reader's understanding.

All figures have been updated and an additional large figure has been added to enhance the reader's understanding.

Reviewer 2 Report

Comments and Suggestions for Authors

This review paper by Farrell et. al. titled “The Current Molecular landscape of Chronic Obstructive Pulmonary Disease (COPD): A review of Diagnoses, Therapies and Efforts Towards Personalized Treatment” shows a nice overview of where the copd field is standing talking about previous, current and emerging therapeutics and diagnosis mechanisms and also talking about molecular mechanisms that may be driving COPD. This is an overall good review; though I have recommended more figures to be added to better depict the complicated cellular and molecular patterns driving COPD as well as main players in COPD development that are relevant to the proteomes readership.

Major comments:

1.     It would be nice to have a schematic of the COPD as a “systemic disease having important bidirectional interactions”.

2.     I think it is important to mention how covid has affected the rates of COPD and in those patients with COPD how covid exacerbated their symptoms.

3.     Please provide a graphical depiction of the following: “Recently we, and many others, have also made critical advances in 123 the immune responses crucial for driving chronic inflammation in COPD that have be-124 come important therapeutic targets [29, 39-44, 48, 49]. These include among many others; 125 mast cells and their proteases, T, B, natural killer (NK) and innate lymphoid cells, macro-126 phages, and neutrophils, inflammasomes, oxidative stress, IL-22, TRAIL, micro(mi)RNAs 127 and the pathways they control, stress granules, amine and lysyl oxidases, and lung and 128 gut microbiomes and the gut-lung axis [29, 42, 50-72]”. For molecular biologists and immunologists this will be important to understand the immune pathways involved in COPD. Also include what is known from adaptive immune cells (T and B cells).

4.     Figure 1 is overly simplified. Please expand more on what molecules are involved inlung inflammation that is part of copd

5.     Create a figure for protease: anti-protease imbalance showing their balance under normal conditions and copd conditions and what enzymes and other factors that are involved with their relative levels.

6.     Create a figure of what you define as the “gold standard” for both diagnosis and treatment of different phenotypes of copd based on the current body of knowledge that we have.

Minor comments:

1.     Please correct all grammatical errors and typos

2.     The references in Table 1 should come in order with the text. There are some references in the 200s but they came before other references in the text. Please fix.

3.     “Smoking rates are declining in some developed countries but continue to increase in 76 others.” Please mention which countries and provide references.

4.     I think that given the current epidemic with e-cigarettes, it is important to devote one entire section to this issue.

5.     Line 89: “indicators of early COPD” please list what these clinical indicators are.

6.     Line 109: “Constant exposure”: Provide a numerical value. Daily exposure, every 5 days, monthly? How is constant exposure defined here.

7.     Line 121: “there five different phenotypes on” should be “there are five different phenotypes on”.

8.     Line 243: List the different eosinophil subtypes and their percent that are found in copd vs asthma. This information is important.

9.     Figure 2: It is a nice depiction of M0 differentiation into M1 or M2s macrophages but what is the connection to COPD? Please add info to the right depicting which macrophage subtypes are differentially increased/decreased in COPD. Also, please add “M unknown” to the figure.

10.  Sections 6, 7, and 9 which are all about drugs should come in order while biomarkers and cellular information of copd should follow.

11.  Line 644: “Experimental data 644 indicated that these specific immune pathways increase airway collagen deposition/fibro-645 sis, but do not influence airway inflammation” What are these pathways, please be more specific and determine exactly which type of T or B cells are involved.

12.  Line 710: “The gut microbiome has been presented as a potential cause 710 for this, with diet and other gut microbiome alterative factors such as antibiotics presenting modulating effects on the immune system, including systemic and airway inflammation”. Which bacterial families are involved or associated with copd? Please be more specific.

13.  Abbreviations should be in alphabetical order.

Comments on the Quality of English Language

Minor grammatical errors and some typos

Author Response

Reviewer 2

This review paper by Farrell et. al. titled “The Current Molecular landscape of Chronic Obstructive Pulmonary Disease (COPD): A review of Diagnoses, Therapies and Efforts Towards Personalized Treatment” shows a nice overview of where the copd field is standing talking about previous, current and emerging therapeutics and diagnosis mechanisms and also talking about molecular mechanisms that may be driving COPD. This is an overall good review; though I have recommended more figures to be added to better depict the complicated cellular and molecular patterns driving COPD as well as main players in COPD development that are relevant to the proteomes readership.

Major comments:

  1. It would be nice to have a schematic of the COPD as a “systemic disease having important bidirectional interactions”.

We understand this comment however we are  emphasizing  a purely molecular/cell-based approach, the reviewer's suggestion would be more appropriate in a paper with a clinical focus  which is not the scope of our manuscript.

  1. I think it is important to mention how covid has affected the rates of COPD and in those patients with COPD how covid exacerbated their symptoms.

While we agree with the importance of recognising the impact Covid has had on COPD, it is our opinion that this would constitute a standalone review, and as such is beyond the scope of our review

  1. Please provide a graphical depiction of the following: “Recently we, and many others, have also made critical advances in 123 the immune responses crucial for driving chronic inflammation in COPD that have be-124 come important therapeutic targets [29, 39-44, 48, 49]. These include among many others; 125 mast cells and their proteases, T, B, natural killer (NK) and innate lymphoid cells, macro-126 phages, and neutrophils, inflammasomes, oxidative stress, IL-22, TRAIL, micro(mi)RNAs 127 and the pathways they control, stress granules, amine and lysyl oxidases, and lung and 128 gut microbiomes and the gut-lung axis [29, 42, 50-72]”. For molecular biologists and immunologists this will be important to understand the immune pathways involved in COPD. Also include what is known from adaptive immune cells (T and B cells).

This has now been adapted to a figure, see figure 1

  1. Figure 1 is overly simplified. Please expand more on what molecules are involved inlung inflammation that is part of copd

This figure has now been updated to better expand upon what molecules are involved in lung inflammation, see figure 2

  1. Create a figure for protease: anti-protease imbalance showing their balance under normal conditions and copd conditions and what enzymes and other factors that are involved with their relative levels.

We appreciate the need for figures to enhance clarity however this does not contribute to the theme of our manuscript and adds a new topic. The scope of our review has been deliberately selected due to the size and possible complexity of the COPD field in general. Again, we feel that this information is in excess of the scope of our manuscript.   

  1. Create a figure of what you define as the “gold standard” for both diagnosis and treatment of different phenotypes of copd based on the current body of knowledge that we have.

As our laboratory group is not made up of clinicians we do not recommend or define any gold standard, we instead defer to the “GOLD report” which is the current clinical guidelines for typing COPD, and the recommendations contained within those guidelines. .

Minor comments:

  1. Please correct all grammatical errors and typos

We have given the paper a thorough read through and have corrected any grammatical errors and typos found.

  1. The references in Table 1 should come in order with the text. There are some references in the 200s but they came before other references in the text. Please fix.

This has now been fixed and came about as part of an endnote error

  1. “Smoking rates are declining in some developed countries but continue to increase in 76 others.” Please mention which countries and provide references.

Regions now listed and WHO referenced, see line 85-86

  1. I think that given the current epidemic with e-cigarettes, it is important to devote one entire section to this issue.

While we recognise that E-Cigarettes are important, research on these are a relatively new field and the connection between them and COPD has not been established yet. We believe this will be a rapidly expanded area in the future but is well beyond the scope of our manuscript.

  1. Line 89: “indicators of early COPD” please list what these clinical indicators are.

This is explained in the next few sentences e.g increased resistance or impaired reactance, to assist with clarity this has been added to lines 100-101.

  1. Line 109: “Constant exposure”: Provide a numerical value. Daily exposure, every 5 days, monthly? How is constant exposure defined here.

We thank the reviewer for this comment, the intended word was chronic exposure as this is generally a pre-defined term that is acceptable to use like this unlike constant exposure. We have changed this in the manuscript. See line 121 https://www.ncbi.nlm.nih.gov/pmc/articles/PMC8035877/

  1. Line 121: “there five different phenotypes on” should be “there are five different phenotypes on”.

Fixed, see line 133

  1. Line 243: List the different eosinophil subtypes and their percent that are found in copd vs asthma. This information is important.

A sentence has been added to clarify that inflammatory Eosinophils are significantly higher and that there are two major types

  1. Figure 2: It is a nice depiction of M0 differentiation into M1 or M2s macrophages but what is the connection to COPD? Please add info to the right depicting which macrophage subtypes are differentially increased/decreased in COPD. Also, please add “M unknown” to the figure.

This has been completed, see Figure 3

  1. Sections 6, 7, and 9 which are all about drugs should come in order while biomarkers and cellular information of copd should follow.

We appreciate this suggestion, however we feel that the  current flow is logical.

  1. Line 644: “Experimental data 644 indicated that these specific immune pathways increase airway collagen deposition/fibro-645 sis, but do not influence airway inflammation” What are these pathways, please be more specific and determine exactly which type of T or B cells are involved.

The pathways are those involving b lymphocytes, t lymphocytes and group 2 innate lymphoid cells , the paper does not differentiate which specific type of cell. https://www.ncbi.nlm.nih.gov/pmc/articles/PMC6487813/

  1. Line 710: “The gut microbiome has been presented as a potential cause 710 for this, with diet and other gut microbiome alterative factors such as antibiotics presenting modulating effects on the immune system, including systemic and airway inflammation”. Which bacterial families are involved or associated with copd? Please be more specific.

The specific bacterial families have now been mentioned, see line 796-799

  1. Abbreviations should be in alphabetical order.

This has now been fixed, see lines 887-935

Reviewer 3 Report

Comments and Suggestions for Authors

The review describes the pathology of COPD, its characters, diagnosis, and treatment with varying degrees of detail. In my opinion, the review lacks a description of its scope in the introduction, which inclusions and exclusion criteria have been chosen for the selected literature. It also lacks a clear objective of the review. As it is now, the reader keep wondering when a statement will come to clarify the aim of the article. Coming to the conclusion, the authors emphasize the use of a specific animal model and does not recognize the importance of in vitro models. That gives the impression of not being objective in describing current research in the field. I recommend reviewing the scope of the article and to define clear objectives to provide a clear direction and purpose for the article, to help readers understand what they can expect to learn or gain from the article.

Comments on the Quality of English Language

I recommend proof-reading the manuscript thoroughly. There are some typing errors, missing words, a few abbreviations that are not introduced, and some concepts that are not described. A few things are described in detailed and some are not and you wonder why it's being mentioned.

Author Response

Reviewer 3

The review describes the pathology of COPD, its characters, diagnosis, and treatment with varying degrees of detail. In my opinion, the review lacks a description of its scope in the introduction, which inclusions and exclusion criteria have been chosen for the selected literature. It also lacks a clear objective of the review. As it is now, the reader keep wondering when a statement will come to clarify the aim of the article. Coming to the conclusion, the authors emphasize the use of a specific animal model and does not recognize the importance of in vitro models. That gives the impression of not being objective in describing current research in the field. I recommend reviewing the scope of the article and to define clear objectives to provide a clear direction and purpose for the article, to help readers understand what they can expect to learn or gain from the article.

While we recognise that the reviewer has strong opinions, we have struggled to find anything actionable for us, we hope that the feedback and actionable requests from reviewers 1, 2 and 4 will satisfy reviewer 3 also.

Reviewer 4 Report

Comments and Suggestions for Authors

This is an extensive review about COPD pathogenesis and implicated pathways. It is informative and well-written.

I have some major comments:

-        Introduction: I think the concept of COPD endotypes is missing here. This concept brought byt the Lancet commission is essential to better understand COPD pathogenesis. It is commonly admitted now that COPD is a complex disorder combining genetic predisposition, early-life events and exposure to inhaled particles and 5 endotypes have been defined according to this new definition.

-        I also miss a sentence concerning the spirometric definition of COPD.

-        Paragraph 6, drugs: other biologics should be briefly cited with corresponding clinical trials (e.g., mepolizumab with METREX/METREO, benralizumab, etc)

-        Paragraph 7, animal models : advantages of animal models are well documented, but drawbacks should be also mentioned in this paragraph (e.g., absence of bronchioles in mouse models).

-        Same paragraph, in vitro models: other types of in vitro models should be mentioned briefly (e.g. bioprinting, organoids, cf https://dx.doi.org/10.1183%2F16000617.0042-2023)

-        Paragraph 8: the title is “molecular landscape” but this paragraph covers mostly cell types (mainly macrophages). Title should be rephrased: cellular landscape instead? And the subparagraphs should be re-organized (ex: grouping all macrophages subtypes). Same remark also for paragraph 9 (where therapies targeting cytokines are put at the same level as microbiota dysregulation or cellular targets).

-        Finally, authors might want to add a figure representing all cellular targets mentioned in COPD lung (or alternatively, complexify Figure 1?)

Minor comments :

-        Paragraph 3, line 127 : I suggest you separate cell types and molecular processes in this sentence (starting from inflammasome) – same remark for line 594.

-        Paragraph 9, subparagraph inflammasomes: is there any history of drugs targeting the inflammasome (such as anakinra) in COPD animal models ?

Typos:

-        Line 136: replace “to COPD” by “in COPD”?

-        Line 148: “ratio” instead of “ration”

-        Line 151: “affected” instead of “effected”

Author Response

Reviewer 4

This is an extensive review about COPD pathogenesis and implicated pathways. It is informative and well-written.

I have some major comments:

-        Introduction: I think the concept of COPD endotypes is missing here. This concept brought byt the Lancet commission is essential to better understand COPD pathogenesis. It is commonly admitted now that COPD is a complex disorder combining genetic predisposition, early-life events and exposure to inhaled particles and 5 endotypes have been defined according to this new definition.

Endotypes introduced now in introduction, see line 63-66

-        I also miss a sentence concerning the spirometric definition of COPD.

Added in introduction based on the GOLD definition , see line 49-50

-        Paragraph 6, drugs: other biologics should be briefly cited with corresponding clinical trials (e.g., mepolizumab with METREX/METREO, benralizumab, etc)

Both biologics mentioned and papers cited, eosinophil count used for inclusion criteria listed. See lines 388-392

-        Paragraph 7, animal models : advantages of animal models are well documented, but drawbacks should be also mentioned in this paragraph (e.g., absence of bronchioles in mouse models).

Absence of bronchioles and reduction of submucosal glands mentioned and referenced. See lines 443-445

-        Same paragraph, in vitro models: other types of in vitro models should be mentioned briefly (e.g. bioprinting, organoids, cf https://dx.doi.org/10.1183%2F16000617.0042-2023)

These are mentioned briefly in the following  section “human ex vivo and in vitro studies” see lines 471-488

-        Paragraph 8: the title is “molecular landscape” but this paragraph covers mostly cell types (mainly macrophages). Title should be rephrased: cellular landscape instead? And the subparagraphs should be re-organized (ex: grouping all macrophages subtypes). Same remark also for paragraph 9 (where therapies targeting cytokines are put at the same level as microbiota dysregulation or cellular targets).

We thank the reviewer for this comment , as we do focus on a number of molecular components alongside cellular we have added cellular to the title. In our opinion the section encompasses both areas equally and hence have amended the title accordingly. -        Finally, authors might want to add a figure representing all cellular targets mentioned in COPD lung (or alternatively, complexify Figure 1?)

We have created a new figure in line with reviewer 1 (see figure 1) that addresses this specific comment.

Minor comments :

-        Paragraph 3, line 127 : I suggest you separate cell types and molecular processes in this sentence (starting from inflammasome) – same remark for line 594.

This has been completed, see lines 133-143 and 663-667

-        Paragraph 9, subparagraph inflammasomes: is there any history of drugs targeting the inflammasome (such as anakinra) in COPD animal models ?

We have added a reference to Daphnetin and Schisandrin A and modified the text  with respect to their specific inflammasome targets in murine models. See lines 733-735

Typos:

-        Line 136: replace “to COPD” by “in COPD”?

This has been done, see line 136

-        Line 148: “ratio” instead of “ration”

This has been done, see line 168

-        Line 151: “affected” instead of “effected”

This has been done, see line 174

Round 2

Reviewer 4 Report

Comments and Suggestions for Authors

Thank you for responding to all my comments. Just a last minor remark: you might want to add "cellular" in the title of paragraph 8 also (in addition to the title of the manuscript).

Author Response

Thank you for your attentiveness i should have caught that

It has now been fixed in the document